# Contrasting Adaptation Mechanisms of Golden *Camellia* Species to Different Soil Habitats Revealed by Nutrient Characteristics

Xianliang Zhu [1], Jianmin Tang [1], Huizhen Qin [1], Kundong Bai [2], Zongyou Chen [1], Rong Zou [1], Shengyuan Liu [3], Quanguang Yang [4], Xiao Wei [1] and Shengfeng Chai [1,*]

1 Guangxi Key Laboratory of Functional Phytochemicals Research and Utilization, Guangxi Institute of Botany, Guangxi Zhuang Autonomous Region and Chinese Academy of Sciences, Guilin 541006, China; xianliangzhu2021@126.com (X.Z.); tjm@gxib.cn (J.T.); qhz0122@sina.com (H.Q.); chenzongyou@gxib.cn (Z.C.); zourong@gxib.cn (R.Z.); weixiao@gxib.cn (X.W.)
2 College of Life Science, Guangxi Normal University, Guilin 541006, China; bkd008@126.com
3 Administration of Nonggang National Nature Reserve of Guangxi, Chongzuo 532400, China; nonggang621@163.net
4 Golden Camellia National Nature Reserve Management Center, Fangchenggang 538001, China; yangquanguang@sina.com
* Correspondence: sfchai@163.com

**Abstract:** Golden *Camellia* species are highly specific to certain soil environments. Most species are only native to calcareous soils in karst regions, except for a few that grow only in acidic soils. Our aim is to elucidate the adaptation mechanisms of the species of calcareous-soil golden *Camellia* (CSC) and acidic-soil golden *Camellia* (ASC) to habitat soils through plant–soil nutrient characteristics and their relationships. We investigated 30 indices for soils and plants. Compared with ASC, CSC had more fertile soil, while their plant tissues exhibited stronger Ca, P, and Mn and weaker K storage, which may be important mechanisms for adapting to habitat soils. However, ASC showed a higher biological absorption coefficient (BAC) for nutrients, which may contribute to the adaptation of ASC to relatively barren acidic soils. Both CSC and ASC showed much higher BAC and accumulation of Ca than other nutrients. We also found that the concentrations of nutrients in the different tissues varied considerably between species. Correlation analysis revealed 135 significant relationships between the 30 indices, with the soil pH and soil Ca levels being the most important factors influencing the nutrient uptake network. This information helps in understanding the adaptation mechanisms of karst plants to habitat soils.

**Keywords:** golden *Camellia* species; karst plant; calcareous soils; acid soils; plant nutrition; adaptability

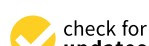



## 1. Introduction

*Camellia* sect. *Chrysantha* Chang, also known as the "the queen of camellias" or "dreaming camellia", is an evergreen shrub or small tree of the Theaceae family that is famous for its golden camellia flowers [1]. These plants primarily grow in Guangxi Province, South China, and North Vietnam [2]. Currently, about 20 species of golden *Camellia* are distributed within China, most of which have a narrow distribution, and all of which are on the List of National Key Protected Wild Plants in China (http://www.gov.cn/ zhengce/zhengceku/2021-09/09/content_5636409.htm, accessed date: 8 November 2021). In natural environments, golden *Camellia* species grow in highly specific soil environments, and most species are only native to calcareous soils, except for a few species that grow only in acidic soils [3]. Depending on their preferred type of soil habitat, they can be divided into calcareous-soil golden *Camellia* (CSC) and acid-soil golden *Camellia* (ASC). A recent cultivation trial showed that CSC also grows normally in low-Ca environments, while ASC is less well adapted to high-Ca environments [4]. The adaptation of these plants to

environments with different Ca ion concentrations may be related to significant variations in photosynthetic and physiological indices, such as their contents of chlorophyll, proline, soluble sugars, or flavonoids [4,5]. However, our knowledge of the mechanisms by which golden *Camellia* species adapt to different soil habitats, especially in the field, is still poor.

Nutrients play key roles in the growth and development of plants, and their profiles can effectively reflect the adaptation mechanisms of plants to specific soil environments [6,7]. C, N, and P are basic elements required by all organisms and are the core elements used to evaluate the nutritional status of plants [8], while certain nutrients, such as K, Ca, Fe, Mg, and Mn, are sources of energy and regulators of many life activities, and thus are considered essential elements in most plants [9]. Nutrients in plants may interact with each other in unexpected ways to maintain the nutritional balance; for example, a deficiency in a single element can lead to the enhanced or reduced uptake of other nutrients by the plant [10]. These nutrients are also frequently used as important indicators for evaluating soil fertility [11,12].

A considerable number of previous studies have investigated plant–soil nutrient characteristics at different scales, i.e., at regional, ecosystem, and species levels [13–15], providing valuable information on the nutrient interactions between plants and soils. Several plants typical of karstic and non-karstic regions have been found to have significantly different leaf Ca contents and Ca storage forms [14]. Cui et al. [15] observed that leaves from a karst forest in Xishuangbanna, Yunnan, and Nonggang, Guangxi, were generally rich in Ca and Mg due to the influence of carbonate rocks. Qi et al. [16] investigated plants of the same genus from different geological backgrounds and revealed that the leaf Ca content of *Primula* growing in karst soils was significantly higher than that of *Primula* in Danxia soils, suggesting that soil type has an important influence on the enrichment of plant nutrients. In golden *Camellia* species, previous studies have documented the nutrient uptake characteristics and habitat soil physicochemical properties of only a few species [17,18]. Nevertheless, the nutrient uptake characteristics of most golden *Camellia* species have not been reported. Moreover, the lack of knowledge about the nutrient interrelationships between plants and soils limits the successful conservation of this group of plants.

As such, in this study, the nutrient profiles of habitat soils, roots, stems, and leaves were analyzed for 14 golden *Camellia* species from calcareous or acidic soils. We aim to explore: (1) What are the differences in nutrients between calcareous and acidic soil habitats? Are these differences in the soils reflected in the concentrations of elements in the plant tissues? (2) What are the differences in nutrient uptake by different golden *Camellia* species? We are particularly interested in the variation in the uptake and storage of Ca by plants. (3) Are there some significant interrelationships in the nutrient exchange between plants and soil? This information will help to reveal the adaptability of golden *Camellia* species to different soil habitats and provide a scientific basis for their conservation.

## 2. Materials and Methods

### 2.1. Plant Materials

The study areas were in Guangxi Zhuang Autonomous Region, China. The 10 species of CSC were *C. impressinervis* (CIM), *C. perpetua* (CPE), *C. longzhouensis* (CLO), *C. pingguoensis var. terminalis* (CPT), *C. flavida* (CFL), *C. huana* (CHU), *C. pubipetala* (CPU), *C. limonia* (CLI), *C. grandis* (CGR), and *C. pingguoensis* (CIM), and the four species of ASC were *C. tunghinensis* (CTU), *C. nitidissima* (CNI), *C. euphlebia* (CEU), and *C. parvipetala* (CPA). Dr. Shengfeng Chai undertook the formal identification of the plant material used in this study. Voucher specimens of this material were deposited in the herbarium of Guangxi Institute of Botany, (voucher number: SFC2017001-SFC2017014). The soil and plant samples were taken from pristine habitats, and the collection sites are shown in Figure 1. The vegetation cover at the collection sites on the northern edge of the tropics was mainly limestone evergreen forest and limestone montane seasonal rainforest, and those in the southern subtropics were evergreen broad-leaved forests with mountain gullies and streams 120–350 m above

sea level, as well as limestone karst slope foothills, crested trough valleys, and depressional valley zones.

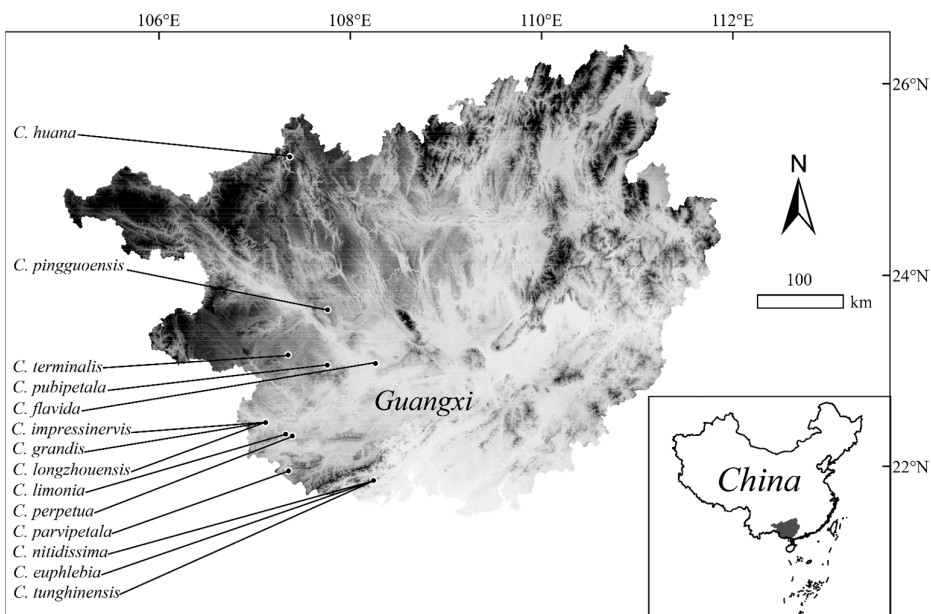

**Figure 1.** Collection sites of 14 species of golden *Camellia*. The map was downloaded from Geospatial Data Cloud (https://www.gscloud.cn, accessed date: 12 October 2021).

### 2.2. Sample Collection and Measurements

One representative population of each species of golden *Camellia* at the collection sites was selected, and three adult plants at a similar growth stage in that population were separately sampled as three replicates. First, about 1 kg of soil was collected from the surface layer (0–20 cm) near the roots of the plant. The soils were then taken to the laboratory, where they were naturally dried and impurities were removed, and they were then ground, sieved, and stored in hermetic bags. In addition, 0.5–1.0 cm of the lateral roots, stems of 1–3 annuals, and leaves of 1 annual were collected from the plants accordingly. These plant samples were taken back to the laboratory in separate clean envelopes, washed, dried, crushed, and then sealed in bags for storage.

A total of 42 soil samples and 126 plant samples were taken. For soil samples, the pH was determined by the glass electrode method; the OM content was determined by the high-temperature external heat potassium dichromate oxidation–volumetric method; the total N and P were determined by a graphite digestion–automatic chemical interruption analyzer; the total K was determined by a graphite digestion–flame photometer; and the total Ca, Mg, Fe, and Mn were determined by microwave digestion–flame atomic absorption spectrophotometry. For the plant samples, the contents of nutrients (N, P, K, Ca, Mg, Fe, and Mn) were determined separately for the root, stem, and leaf samples using the same method as described above. A total of 30 different indices were measured in the soil and plant samples. To effectively quantify the absorption capacity of the plants for each nutrient in the soil, the biological absorption coefficient (BAC) for each nutrient was calculated with reference to the equation of de la Fuente et al. [19]:

$$BAC_E = [(C_{PR} + C_{PS} + C_{PL})/3]/C_S \qquad (1)$$

where $C_{PR}$, $C_{PS}$, and $C_{PL}$ are the element's (E) mean concentrations in the root, stem, and leaf parts of the plants expressed in g kg$^{-1}$ d.w., and $C_S$ is the element's mean concentration in soil expressed in g kg$^{-1}$. Typically, BAC values in the range of 0.01–0.1, 0.1–1, and 1–10 indicate weak, moderate, and strong absorption, respectively.

### 2.3. Statistical Analysis

R v4.1.1 [20] was used for statistical analysis. First, all indices were tested for normality with the Shapiro–Wilk test at $\geq 0.9$; when necessary, the indices were subjected to mathematic transformations (e.g., logarithm, sine, or cosine). Then, a *t*-test was performed to compare the differences between the indices of the soil, roots, stems, and leaves of CSC and ASC. Differences in the nutritional characteristics (N, P, K, Ca, Mg, Fe, and Mn in roots, stems, and leaves) between species were compared using ANOVA followed by Duncan's multiple comparison test ($p < 0.05$). The nutrient characteristics in the roots, stems, and leaves were clustered using the "ward.D" method [21]. Analyses of the correlations between 30 indices were tested by Pearson's correlation coefficient. Significant correlation coefficients were extracted using a threshold of $p < 0.05$, followed by the construction of relationship network plots using Cytoscape v3.8.2 [22]. Heat maps were drawn using TBtools v1.098669 [23].

### 3. Results

#### 3.1. Differences in Habitat Soil between CSC and ASC

The soil pH preferred by CSC species ranged from 6.61 (CIM) to 7.53 (CPI), while the soil pH of ASC species ranged from 3.81 (CNI) to 5.86 (CPA) (Table 1). The mean soil organic matter (OM) content for calcareous soil (8.05%) was significantly higher ($p < 0.05$) than that for acidic soil (5.95%). The content of each nutrient was ranked as Fe > Mg > K > Ca > N > Mn > P for calcareous soils, and Fe > K > Mg > N > Ca > P > Mn for acidic soils. Except for the K content, all other nutrients were present at higher concentrations in calcareous soils than in acidic soils.

**Table 1.** The pH, organic matter (OM), and major nutrient contents of the habitat soils of golden *Camellia* species.

| Species | Soil-pH | Soil-OM (%) | Soil-N (g kg$^{-1}$) | Soil-P (g kg$^{-1}$) | Soil-K (g kg$^{-1}$) | Soil-Ca (g kg$^{-1}$) | Soil-Mg (g kg$^{-1}$) | Soil-Fe (g kg$^{-1}$) | Soil-Mn (g kg$^{-1}$) |
|---|---|---|---|---|---|---|---|---|---|
| CIM ($n = 3$) | 6.61 ± 0.48 b | 6.30 ± 1.76 cd | 4.03 ± 1.64 bc | 1.88 ± 0.21 ab | 9.87 ± 1.91 a | 2.54 ± 0.94 de | 6.35 ± 1.44 cde | 105.83 ± 8.89 a | 2.19 ± 0.15 c |
| CPE ($n = 3$) | 7.00 ± 0.51 ab | 5.58 ± 0.72 cd | 1.56 ± 0.37 ef | 0.54 ± 0.06 e | 1.00 ± 0.30 h | 3.31 ± 0.96 bcd | 7.31 ± 0.13 cd | 90.00 ± 1.99 b | 1.20 ± 0.05 de |
| CLO ($n = 3$) | 6.93 ± 0.25 ab | 9.17 ± 0.89 b | 2.96 ± 0.22 cde | 2.03 ± 0.27 a | 2.35 ± 0.10 gh | 5.12 ± 1.01 ab | 9.93 ± 0.46 b | 86.39 ± 5.66 bc | 2.70 ± 0.37 bc |
| CPT ($n = 3$) | 6.90 ± 0.35 ab | 12.31 ± 2.69 a | 6.69 ± 1.64 a | 0.74 ± 0.04 de | 2.33 ± 0.29 gh | 1.95 ± 0.76 e | 7.95 ± 0.20 c | 80.20 ± 3.29 bcd | 1.32 ± 0.19 d |
| CFL ($n = 3$) | 7.21 ± 0.21 ab | 7.29 ± 0.83 c | 2.21 ± 0.50 cdef | 1.78 ± 0.20 bc | 6.65 ± 0.97 bcd | 3.53 ± 1.26 bc | 5.75 ± 0.44 de | 88.44 ± 1.79 bc | 4.31 ± 1.16 a |
| CHU ($n = 3$) | 7.19 ± 0.37 ab | 6.95 ± 1.66 c | 2.68 ± 0.53 cdef | 1.06 ± 0.04 d | 8.03 ± 1.09 b | 4.50 ± 1.48 ab | 3.83 ± 1.36 f | 52.17 ± 10.46 f | 3.50 ± 0.80 ab |
| CPU ($n = 3$) | 7.28 ± 0.13 ab | 9.87 ± 1.60 b | 3.06 ± 0.50 cde | 1.69 ± 0.33 bc | 4.63 ± 0.56 def | 5.96 ± 1.49 a | 13.87 ± 1.12 a | 88.26 ± 6.91 bc | 2.91 ± 0.23 bc |
| CLI ($n = 3$) | 6.93 ± 0.17 ab | 6.46 ± 1.21 cd | 3.85 ± 1.42 bcd | 0.85 ± 0.07 de | 4.5 ± 0.1 ef | 3.49 ± 0.57 bc | 3.68 ± 0.08 f | 82.72 ± 2.76 bc | 2.75 ± 0.46 bc |
| CGR ($n = 3$) | 6.98 ± 0.59 ab | 6.62 ± 0.67 c | 5.05 ± 1.00 b | 1.07 ± 0.33 d | 3.87 ± 0.11 fg | 5.88 ± 1.62 a | 7.89 ± 0.92 c | 70.09 ± 10.52 de | 2.81 ± 0.74 bc |
| CPI ($n = 3$) | 7.53 ± 0.28 a | 9.94 ± 1.10 b | 2.17 ± 1.16 def | 1.50 ± 0.20 c | 3.10 ± 0.89 fg | 5.38 ± 1.36 a | 9.75 ± 1.23 b | 77.25 ± 2.16 cd | 2.36 ± 0.60 c |
| **CSC ($n = 30$)** | **7.06 ± 0.25 \*\*** | **8.05 ± 2.15 \*\*** | **3.43 ± 1.54 \*\*** | **1.31 ± 0.53 \*\*** | **4.63 ± 2.78 \*\*** | **4.16 ± 1.41 \*\*** | **7.63 ± 3.06 \*\*** | **82.13 ± 14.10 \*\*** | **2.61 ± 0.93 \*\*** |
| CTU ($n = 3$) | 5.00 ± 0.37 d | 8.01 ± 0.14 bc | 2.11 ± 0.35 def | 0.76 ± 0.02 de | 7.1 ± 1.31 bc | 0.60 ± 0.12 e | 6.62 ± 1.12 cde | 50.71 ± 2.30 f | 0.50 ± 0.10 de |
| CNI ($n = 3$) | 3.81 ± 0.23 e | 5.59 ± 1.44 cd | 3.24 ± 0.94 cde | 0.64 ± 0.03 e | 6.07 ± 0.61 cde | 0.40 ± 0.16 e | 3.85 ± 1.11 f | 51.20 ± 4.18 f | 0.33 ± 0.07 e |
| CEU ($n = 3$) | 5.21 ± 0.57 d | 6.20 ± 0.95 cd | 2.18 ± 0.27 def | 0.73 ± 0.06 de | 9.83 ± 1.31 a | 0.48 ± 0.10 e | 6.34 ± 1.87 cde | 61.19 ± 2.89 ef | 0.63 ± 0.23 de |
| CPR ($n = 3$) | 5.86 ± 0.15 c | 3.97 ± 1.06 d | 1.07 ± 0.18 f | 0.56 ± 0.10 e | 7.47 ± 1.98 bc | 1.47 ± 0.33 e | 5.15 ± 1.30 ef | 29.99 ± 7.55 g | 0.73 ± 0.41 de |
| **ASC ($n = 12$)** | **4.97 ± 0.86 \*\*** | **5.94 ± 1.69 \*\*** | **2.15 ± 0.89 \*\*** | **0.67 ± 0.09 \*\*** | **7.62 ± 1.59 \*\*** | **0.74 ± 0.50 \*\*** | **5.49 ± 1.26 \*\*** | **48.27 ± 13.11 \*\*** | **0.55 ± 0.17 \*\*** |

Data are the mean ± standard deviation. Different letters after the data in the same column indicate significant differences ($p < 0.05$) after Duncan's multiple comparison test. The bold text represents the average values of each index of CSC and ASC, and \*\* indicates extremely significant differences ($p < 0.01$) between CSC and ASC according to the *t*-test.

#### 3.2. Differences in Major Nutrients Contents and Biological Absorption Coefficients between CSC and ASC

Eleven nutrient indices (mainly P, K, and Ca) of the plants differed significantly ($p < 0.05$) between CSC and ASC, eight of which were observed in the aerial parts (stems and leaves) (Table 2). P and Ca were significantly higher in the roots, stems, and leaves of CSC than in ASC, while the opposite was true for K. Fe and Mn differed significantly ($p < 0.05$) only in the stems. In terms of the nutrient uptake capacity, all species had the highest BAC$_{Ca}$ and the lowest BAC$_{Fe}$ (Figure 2). Among the species, CEU had the highest cumulative BAC values for the seven elements, while CPU had the lowest. CPE showed the highest BAC$_N$ and BAC$_K$. As a whole, CSC showed lower BAC values of most elements compared with ASC.

**Table 2.** Nutrient differences in the roots, stems, and leaves of CSC and ASC.

| Indices (g kg$^{-1}$) | CSC (*n* = 30) | ASC (*n* = 12) |
| --- | --- | --- |
| Root-N | 4.46 ± 2.31 NS | 5.72 ± 2.79 NS |
| Root-P | 0.89 ± 0.52 ** | 0.53 ± 0.10 ** |
| Root-K | 2.45 ± 1.48 * | 3.46 ± 1.593 * |
| Root-Ca | 17.60 ± 9.13 * | 11.16 ± 11.31 * |
| Root-Mg | 2.30 ± 1.13 NS | 2.62 ± 1.09 NS |
| Root-Fe | 0.39 ± 0.30 NS | 0.31 ± 0.11 NS |
| Root-Mn | 0.75 ± 0.70 NS | 0.71 ± 0.54 NS |
| Stem-N | 4.39 ± 2.10 NS | 4.01 ± 1.15 NS |
| Stem-P | 0.93 ± 0.39 ** | 0.62 ± 0.08 ** |
| Stem-K | 2.76 ± 1.33 * | 4.32 ± 1.94 * |
| Stem-Ca | 22.98 ± 10.33 ** | 6.62 ± 2.03 ** |
| Stem-Mg | 3.06 ± 1.55 NS | 2.46 ± 1.42 NS |
| Stem-Fe | 0.30 ± 0.20 * | 0.43 ± 0.24 * |
| Stem-Mn | 0.98 ± 0.87 ** | 0.32 ± 0.08 ** |
| Leaf-N | 8.65 ± 5.63 NS | 7.12 ± 3.85 NS |
| Leaf-P | 0.84 ± 0.30 ** | 0.61 ± 0.14 ** |
| Leaf-K | 6.46 ± 2.63 ** | 10.62 ± 4.32 ** |
| Leaf-Ca | 20.59 ± 10.14 * | 13.45 ± 7.22 * |
| Leaf-Mg | 2.26 ± 0.84 NS | 2.40 ± 1.41 NS |
| Leaf-Fe | 0.28 ± 0.16 NS | 0.29 ± 0.11 NS |
| Leaf-Mn | 1.49 ± 1.42 NS | 0.77 ± 0.37 NS |

Data are the mean ± standard deviation. * and ** indicate significant differences between CSC and ASC at the $p$ = 0.05 and $p$ = 0.01 levels, respectively, and NS indicates no significant difference.

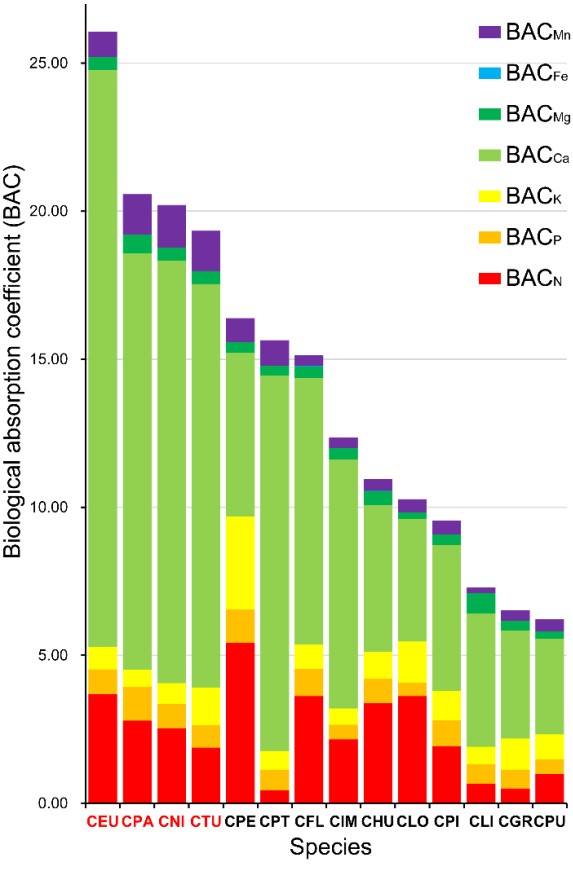

**Figure 2.** Biological absorption coefficient of each nutrient in golden *Camellia* species. The right subscripts of BAC in the legend represent the different elements. The black and red species labels represent CSC and ASC, respectively.

### 3.3. Nutrient Characteristics of Golden Camellia Species

The one-way analysis of variance (ANOVA) of the nutrient contents of the roots, stems, and leaves of 14 species of golden *Camellia* showed extremely significant ($p < 0.01$) differences in all 20 indices except for the leaf Fe content ($p < 0.05$) (Figure 3). In most species, the accumulation of N in the leaves was much higher than that in the stems and roots, with the leaf N content being the highest in CLO, at 18.01 g kg$^{-1}$. We found that two species (CFL and CPI) showed a strong preference for P, as both had considerably higher P contents in their roots, stems, and leaves than the other species. The K content formed a similar pattern to N, with a higher content in the leaves than the stems and roots in most species. However, the Ca content showed greater divergence between species, with most of the CSC plants exhibiting a decrease in Ca in the order of stems, leaves, and roots, while ASC exhibited a decrease in the order of leaves, roots, and stems. Mg mainly accumulated in the stems of most species, including CPT, CGR, CLO, and CLI, and in the leaves in a few species, such as CTU and CPE. All species displayed higher Fe contents in the aerial parts than the roots, except for CPU. The distribution of Mn in golden *Camellia* species showed remarkable variability; for example, Mn was mainly concentrated in the roots of CPI, CPT, and CPA; in the stems of CLO, CGR, and CIM; and in the leaves of CHU, CFL, CPU, CPE, and CTU. Cluster analysis showed that CTU, CTE, and CNI had similar elemental uptake characteristics in the roots, stems, and leaves, while CPA was close to other CSCs (especially CPT). In most species, the concentration of Ca was much higher than that of other nutrients, indicating that Ca is an important plant nutrient in golden *Camellia* species.

### 3.4. Plant–Soil Nutrient Relationships of Golden Camellia Species

A total of 135 significant ($p < 0.05$) relationships were detected between 30 soil and plant indices (Figure 4). Among them, the highest number of significant relationships (16) was found between soil pH and other indices, followed by soil Ca (15), indicating that the pH and Ca content of soil have the most important effects on nutrient absorption by golden *Camellia* plants. There were 13 and 11 significant relationships between the soil P and Mn and other indices, respectively, such as between the soil P and root P ($R = 0.54$), stem P ($R = 0.75$) and leaf P ($R = 0.60$), and soil Mn and leaf Mn ($R = 0.49$) and stem Mn ($R = 0.32$), further indicating the significant effects of the soil habitat on nutrient absorption. When the relationships among the plant indices were assessed, stem Mg had the highest number of significant relationships (13) with the other indices, while the leaf Fe had the lowest, i.e., one significantly negative relationship with leaf Mn ($R = -0.31$), indicating that stem Mg is susceptible to the influence of other nutrients, while the opposite is true for leaf Fe. There were strong positive relationships between the three essential nutrients N, P, and K in different tissues of golden *Camellia* species; for example, leaf N and stem N ($R = 0.86$), leaf N and root N ($R = 0.78$), and stem N and root N ($R = 0.68$), suggesting that the amount of absorption of these essential elements by other tissues of the plant is reflected in the leaves. In contrast, there were negative associations between the Mg contents of different tissues; for example, leaf Mg and stem Mg ($R = -0.32$). In addition, we found significant synergistic effects (for example, the correlation coefficients for the relationship between Ca and Mn were as high as 0.71, 0.73, and 0.70 in the roots, stems, and leaves, respectively) and antagonistic effects (for example, root N and root Ca ($R = -0.44$) and root Mn ($R = -0.44$), stem Fe and stem Mn ($R = -0.42$), and stem K and stem Mg ($R = -0.43$)) on the absorption of several elements.

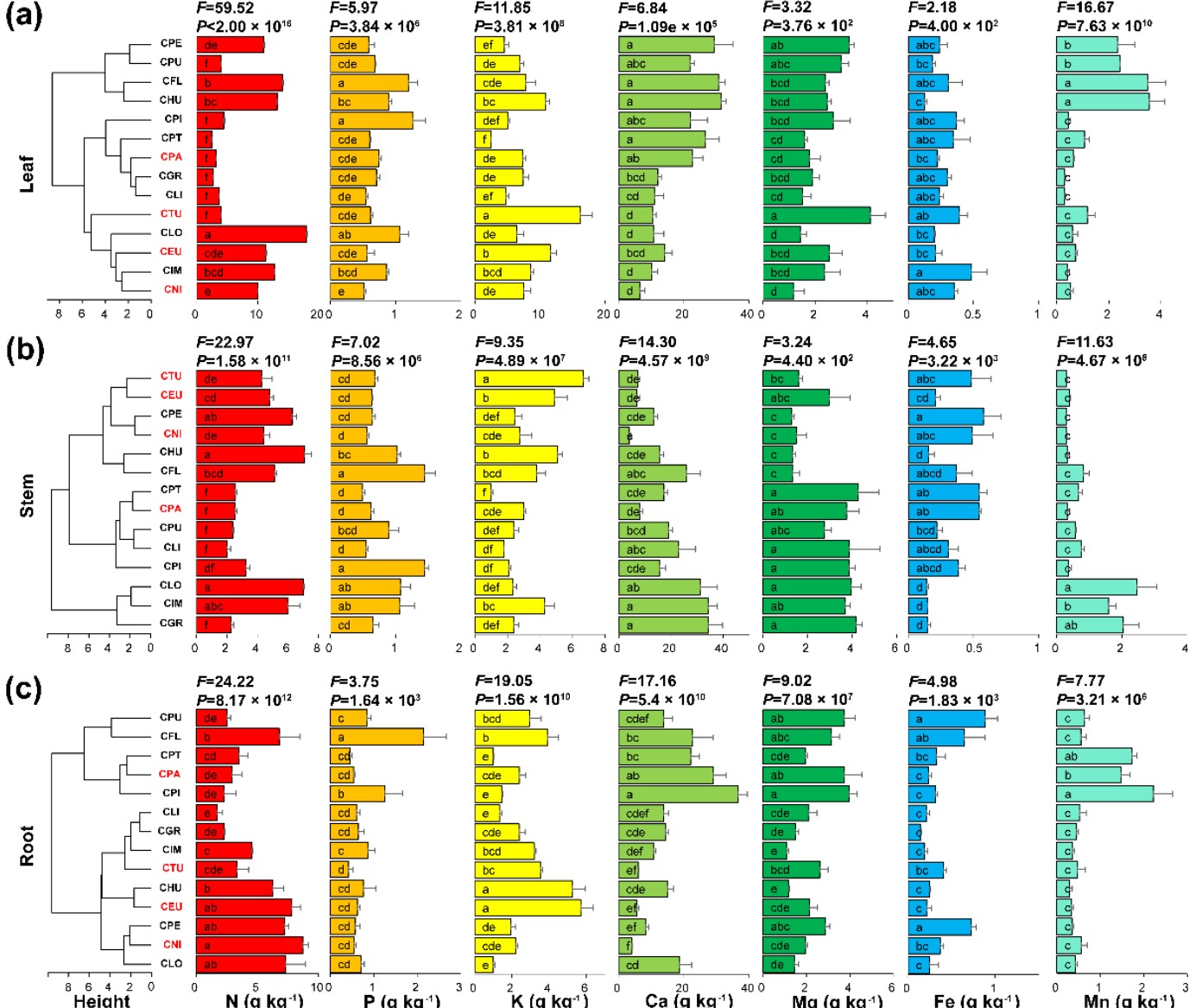

**Figure 3.** Comparison of the N, P, K, Ca, Mg, Fe, and Mn contents among leaves (**a**), stems (**b**), and roots (**c**) of 14 golden *Camellia* species. The black and red species labels at the ends of the clustering trees represent CSC and ASC, respectively. F and P are the statistics and significance of the ANOVA, respectively. The lines in the bar chart represent the standard deviation. Different letters indicate significant differences (*p* < 0.05) according to Duncan's multiple comparison test.

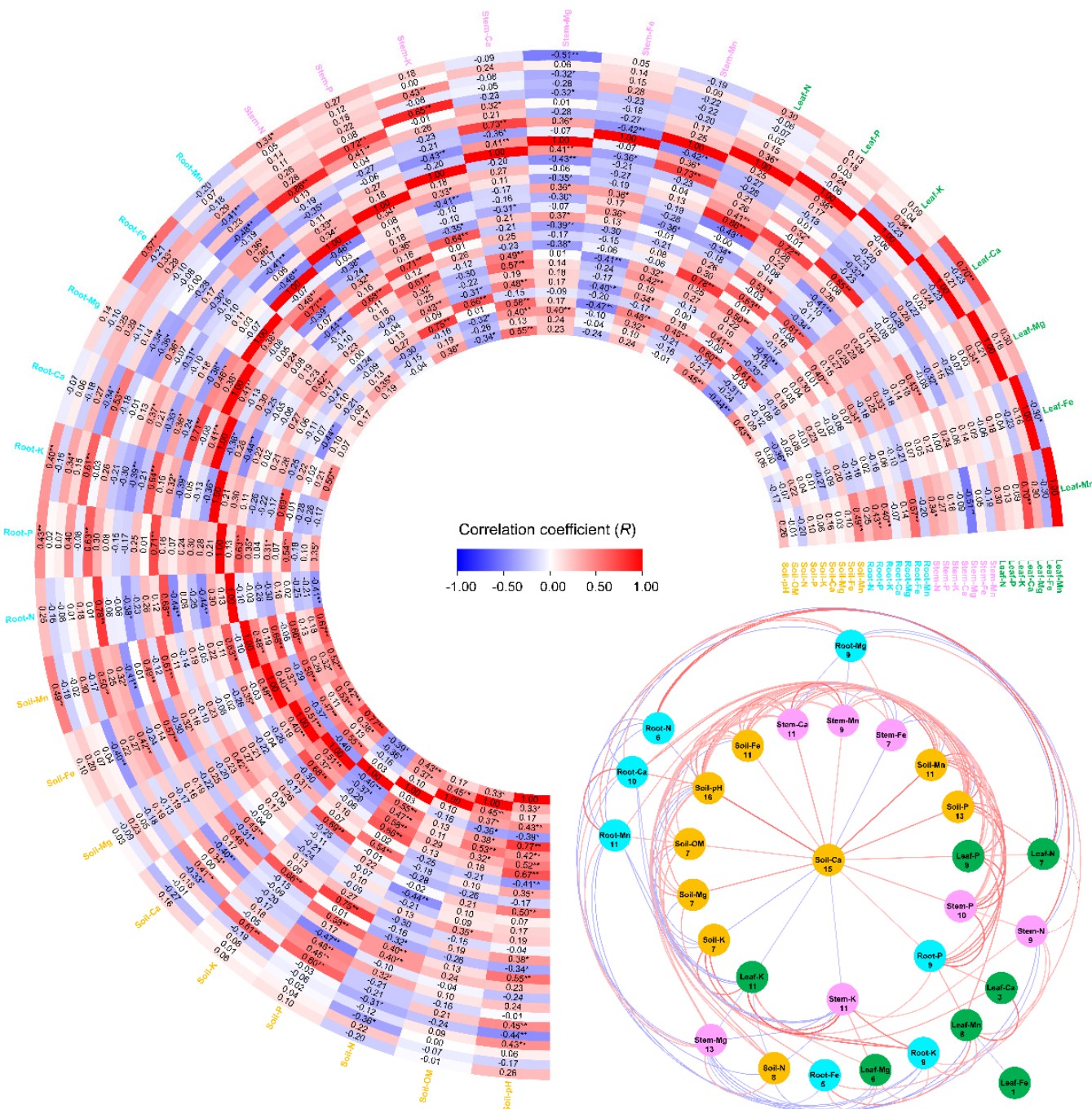

**Figure 4.** Plant–soil nutrient relationships of golden *Camellia* species. The heat map shows the correlation coefficients between 30 indices of soil and plants. * and ** indicate significant correlation coefficients at the $p = 0.05$ and $p = 0.01$ levels. Networks were constructed by extracting significant correlation coefficients from the heat map ($p < 0.05$). The circles indicate indices, their colors correspond to the indexes on the heat map, and the values underneath each index indicate the number of connected nodes. The color of the lines between the circles represents the correlation coefficient.

## 4. Discussion

Previous studies highlighted the higher pH and Ca content of calcareous soils in comparison with acidic soils [14,24]. Our results support the previous studies and, in addition, further confirmed that calcareous soils were enriched with higher contents of OM and most nutrients, such as N (3.45 g kg$^{-1}$), P (1.31 g kg$^{-1}$), and Mg (7.63 g kg$^{-1}$) (Table 1). This may be attributed to the abundance of Ca in the bedrock of karstic calcareous soils.

Elemental Ca is reported to have adsorption and precipitation effects on other nutrients, thus allowing for better nutrient fixation in the soil [25]. The soil bedrock of acidic soils is mostly sand shale with a low Ca content, which, together with the acidic soil environment, exacerbates Ca and Mg leaching [26] and promotes the decomposition of OM [27], which may lead to a further decrease in soil fertility. In addition, different topographies, soil textures, weathering environments, and other factors may also lead to differences in soil nutrients [28]. For example, both the N content and P content of CSC soils were higher than those previously observed in karst limestone soils in northwestern Guangxi [29]. Our results suggest that CSC colonized a relatively more fertile habitat than ASC.

The differences in the nutrient elements in soils may directly lead to different distributions of the corresponding nutrient elements in plants [30]. Thus, the ability of plants to store soil nutrients can serve as an adaptive response to soil nutrient variability [31]. The differences in Ca, P, Mn, and K enrichment between CSC and ASC may be a direct result of differences in the amount of nutrients in the habitat soils (Tables 1 and 2), suggesting that golden *Camellia* species are more sensitive to variations in these elements in the soil. Conversely, although the contents of several other nutrients, such as N, Fe, and Mg, also differed considerably between limestone and acidic soils, significant differences in enrichment were not observed in the nutrient tissues of most species (Tables 1 and 2), suggesting that the uptake of this class of nutrients by golden *Camellia* species is weakly regulated by the soil environment. In most species, the accumulation of N, P, and K elements by leaves is dominant, compared with stems and roots (Figure 3). This may be due to leaves being a vital organ for plants to carry out vigorous metabolic functions requiring many basic elements [32,33]. However, the accumulation of several nutrients, especially Ca and Mn, showed significant differences between tissues and species (Figure 3). In regions of high habitat heterogeneity, plants adapt to specific conditions by adjusting their nutrient and water uptake, biomass, spatial distribution characteristics, and morphological structure [34]. Different golden *Camellia* species may be better adapted to karst or non-karst areas with high habitat heterogeneity by regulating nutrient accumulation between different tissues. For instance, the difference in Mn accumulation between CSC and ASC was mainly observed in the stem, while the difference in K accumulation was mainly observed in the leaves (Table 2). Thus, a key insight is to focus on integrating the nutritional contents of different tissues when studying the plants' nutritional characteristics in these highly heterogeneous habitats [35].

The nutrient uptake of plants in karst or non-karst habitats is thought to be related to their calcicole or calcifuge behavior [24]. Therefore, we are particularly interested in the Ca absorption characteristics of golden *Camellia* species. The Ca uptake efficiency and accumulation of golden *Camellia* species were much higher than those of other nutrients (Figures 2 and 3), implying that soil Ca is one of the most important elements affecting growth and development. Depending on their Ca requirements, plants can be classified as calciphile, calcicole, calcifuge, sub-calcifuge, or neutral [36]. Calcicoles, in particular, are characterized by their ability to grow normally in high-Ca soils and are rarely found in acidic soils, while calcifuges grow well in acidic soils but are harmed by a slight increase in the soil Ca content. Calcicoles generally have a greater capacity for Ca uptake and storage than calcifuges [14–16,35]. In fact, CSC and ASC are calcicole and calcifuge plants, respectively. The study of calcicole or calcifuge behavior in plants remains a hot topic; however, it has rarely been discussed at the genus or species scales. Luo et al. [37] found that most typical calcicoles in karst forests were characterized by low P and K and high Ca and Mg, and most are P-limited plants (N/P > 16). However, CSC exhibited high P and similar Mg levels to ASC, except for the K and Ca uptake characteristics consistent with typical calcicoles. Furthermore, the N/P values in the roots, stems, and leaves of CSC were all lower than 14, indicating that they were N-limited, rather than P-limited [37]. These elemental uptake characteristics may reveal a unique mechanism for the adaptation of golden *Camellia* species to different soil habitats.

In terms of the uptake efficiency of different nutrients, golden *Camellia* species had a low BAC for Fe, the most abundant element in the soil, but a high BAC for other, less abundant nutrients (Figure 2). Similar nutrient absorption characteristics have been reported in previous studies of plants, such as several dominant species of *Burretiodendron hsienmu*, *Litsea dilleniifolia*, and *Cephalomappa sinensis* in the tropical and subtropical karst forest regions of Guangxi [38], as well as the endemic limestone species *Triadica rotundifolia* [39]. Interestingly, ASC has a higher BAC than CSC for most nutrients. This suggests that the more efficient nutrient uptake efficiency may allow ASC to adapt to more infertile acidic soil habitats.

The soil pH and soil Ca content have significant effects on plant nutrient uptake, growth, and development by influencing the physical, chemical, and biological properties of the soil [39,40]. We further confirmed that these two important soil indices have the most extensive effects on plants by constructing a network of relationships between the major nutrient contents of *Camellia* spp. and soil (Figure 4). The plant index with the highest correlation with both the soil pH and soil Ca was Stem-Ca, indicating that the Ca content in the stems of golden *Camellia* species would be most affected if there were changes in the soil pH or Ca. Therefore, it is necessary to focus on the soil pH and soil Ca content in the future cultivation or conservation of golden *Camellia* species. Additionally, we identified a few significant relationships between the nutrient elements. The relationships between some nutrients were mentioned in previous reports; for example, the synergistic effects of N and P [40]. However, the interactions between some nutrients in this study were inconsistent with previous reports. For instance, in *Brassica napus*, an elevated Mn content had a significant inhibitory effect on the absorption of both Ca and Fe [41]; however, in golden *Camellia* species, Mn showed a significant synergistic effect with Ca. On the one hand, this may be due to the unique biological properties distinguishing the species. On the other hand, synergistic or antagonistic interactions between nutrients in plant–soil systems may be closely related to their concentrations [42]. The absorption of K and Mg, as an example, was synergistic when the soil K content was low, and antagonistic when the soil K content was high [43]. It is worth noting that K and Mg showed antagonistic effects in this study, suggesting that the K content of the habitat soil was more than adequate for golden *Camellia* species. Nevertheless, the nutrient interrelationships between plants and soils need to be further verified in conjunction with physiological experiments.

## 5. Conclusions

By investigating the nutrient characteristics of the habitat soils and plant tissues of 14 species of golden *Camellia*, the following conclusions were drawn: (1) neutral to weakly alkaline calcareous soil habitats are more fertile than acidic soil habitats. (2) CSC exhibited stronger Ca, P, and Mn and weaker K storage than ASC, yet ASC had higher nutrient uptake efficiency, suggesting their contrasting mechanisms of adaptation to habitat soils. (3) A complex soil–plant nutrient exchange network was found, in which soil Ca and soil pH play important roles. Overall, this study provides a scientific basis for the conservation of the germplasm resources of this rare species by revealing the adaptation of plants of the golden *Camellia* species to different types of soils from the perspective of plant nutrition.

**Author Contributions:** Conceptualization, X.W. and S.C.; Data curation, X.Z. and J.T.; Formal analysis, X.Z. and S.C.; Funding acquisition, X.W. and S.C.; Investigation, J.T., H.Q., Z.C., R.Z., S.L., Q.Y. and S.C.; Project administration, R.Z.; Resources, S.L., Q.Y. and X.W.; Software, X.Z.; Visualization, X.Z.; Writing—original draft, X.Z.; Writing—review and editing, K.B. and S.C. All authors have read and agreed to the published version of the manuscript.

**Funding:** This research was supported by the National Natural Science Foundation of China (32060248, 31660092, 31860169), the key research and development program of Guangxi (GuikeAB21196018), the Central Guidance on Local Science and Technology Development Fund (ZY21195035), and Yunfu City 2021 Chinese medicine (southern medicine) industry talent project (Yunke [2022] No.16).

**Institutional Review Board Statement:** Not applicable.

**Informed Consent Statement:** Not applicable.

**Data Availability Statement:** Not applicable.

**Acknowledgments:** We thank the Golden Camellia National Nature Reserve Management Center, the Administration of Nonggang National Nature Reserve of Guangxi, and the Guangxi Longhushan Nature Reserve Management Office for their help in sample collection.

**Conflicts of Interest:** The collection of plants and soils of endangered golden *Camellia* species in this study was granted by the Golden Camellia National Nature Reserve Management center, the Administration of Nonggang National Nature Reserve of Guangxi, and the Guangxi Longhushan Nature Reserve Management Office, and the collection of these materials complied with the Regulations of the People's Republic of China on the Protection of Wild Plants and the IUCN Policy Statement on Research Involving Species at Risk of Extinction. The authors declare that they have no competing interests.

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
