# Peer review of "Contrasting Adaptation Mechanisms of Golden Camellia Species to Different Soil Habitats Revealed by Nutrient Characteristics"

_agronomy, doi:10.3390/agronomy12071511_

Round 1

Reviewer 1 Report

The work presented by Zhu et al. reveals interesting connections between adaptive mechanisms and soil composition in a well-structured dataset of Camellia species/habitats.

All the parts of this work support the experimental design and the conclusions reported by the authors. 

I have no comments to add.

Author Response

Point 1: The work presented by Zhu et al. reveals interesting connections between adaptive mechanisms and soil composition in a well-structured dataset of Camellia species/habitats. All the parts of this work support the experimental design and the conclusions reported by the authors. I have no comments to add.

Response1: We appreciate the positive remarks of this reviewer.

Reviewer 2 Report

The manuscript is well prepared. The visualisation is elegantly presented.

Author Response

Point 1The manuscript is well prepared. The visualisation is elegantly presented.

Response1 : We thank the reviewer for the recognition of our work in visualization.

Reviewer 3 Report

Need for this project has been poorly described. A clear hypothesis and what this research work adds to the previous knowledge should be given in the last paragraph of Introduction. Moreover, it should be updated with latest papers from 2021-22

Zhao, J., Liu, T., Zhang, D., Wu, H., Zhang, T., Dong, D. and Liao, N., 2021. Bacterial Community Composition in the Rhizosphere Soil of Three Camellia chrysantha Cultivars Under Different Growing Conditions in China. Journal of Soil Science and Plant Nutrition, 21(4), pp.2689-2701.

Discussion is very weak and shallow, it should be more deep focusing on the responsible mechanisms. Moreover, it should be updated with latest papers from 2022

The quality of Figure is poor. It legend is difficult to read. It should be improved. Also need to be described in the caption of the Figure

In the whole paper, there is not any mention of the replicates used during sampling and analysis. These must be provided in materials and Methods section, with Figure and Tables where required 

Author Response

Point 1: Need for this project has been poorly described. A clear hypothesis and what this research work adds to the previous knowledge should be given in the last paragraph of Introduction. Moreover, it should be updated with latest papers from 2021-22.

Zhao, J., Liu, T., Zhang, D., Wu, H., Zhang, T., Dong, D. and Liao, N., 2021. Bacterial Community Composition in the Rhizosphere Soil of Three Camellia chrysantha Cultivars Under Different Growing Conditions in China. Journal of Soil Science and Plant Nutrition, 21(4), pp.2689-2701.

Response1: We added some previous studies in the Introduction (line 47-50, 73-74). To our knowledge, few studies on nutrient characteristics have been carried out in golden Camellia species in the field. In addition, the nutrient characteristics and nutrient interrelationships of most golden Camellia species are still unknown, limiting the successful conservation of such plants. Therefore, our study will serve as a well complement to the previous knowledge. A clear hypothesis of this study is that nutrient differences exist between soils and plants in calcareous soil and acidic soil habitats. However, what nutrient differences exist that we do not know (line 81-83). In the Introduction, we replaced the most recent papers starting from 2022 as much as possible. For example, Zhao et al. 2021, Liu et al. 2022.

Point 2: Discussion is very weak and shallow, it should be more deep focusing on the responsible mechanisms. Moreover, it should be updated with latest papers from 2022

Response2: We made some changes to the Discussion. We summarize the nutrient characteristics of golden Camellia species in calcareous soil habitats, specifically high Ca, P, Mn and weaker K storage capacity. In relatively infertile acidic soil habitats, golden Camellia species showed more efficient nutrient uptake. These may be important mechanisms for their adaptation to different habitats. In addition, we consider that the interactions between nutrients may have important reference value for the subsequent cultivation and conservation of golden Camellia species. Therefore, some interesting nutrient interactions are also discussed. As suggested by the reviewer, we also updated some new papers in the Discussion.

Point 3: The quality of Figure is poor. It legend is difficult to read. It should be improved. Also need to be described in the caption of the Figure.

Response3: We added some descriptions to the caption of Figure 2 for easy reading (line 177-178). In Figure 3, the standard deviation lines and the corresponding descriptions were added (line 205). The colors of the circles in the network diagram in Figure 4 correspond to the indexs in the heat map for easy understanding, and some descriptions were also added (line 236-238).

Reviewer 4 Report

The ms shows low originality, since most of the parts (e.g. the Dicsussion about calcifuge and calciphile plants) have been already discussed in the past; in addition, although nutrient interrelations are interesting, most of the data are already known (e.g. Mg/K interactions e.t.c.) from previous studies.  

Finally, the way of some data presentation needs improvement, since the way of presentation is monotonous (e.g. Figure 3).   

Author Response

Point 1: The ms shows low originality, since most of the parts (e.g. the Dicsussion about calcifuge and calciphile plants) have been already discussed in the past; in addition, although nutrient interrelations are interesting, most of the data are already known (e.g. Mg/K interactions e.t.c.) from previous studies. 

Response1: To highlight the originality of our study, we have added some descriptions (line 50-52, 73-76, 290-291). Although much work has been done in the past on calcifuge and calciphile plants, they have rarely been discussed at the species or genus scale. In Discussion, we cite some results that are consistent with previous studies, suggesting that CSC and ASC have some prevalence of calcifuge and calciphile plants, and also validate the reliability of our results. However, our results are not in full agreement with previous studies, such as the description of lines 290-270. In addition, the interrelations among nutrients are complex and have different performances in different species or in different treatments. It needs to be explored by spending a lot of large indoor experiments and field work. Our study emphasizes the nutrient interrelations among golden Camellia species in their original habitats. These nutrient interrelations are important for the future cultivation and conservation of golden Camellia species.

Point 2: Finally, the way of some data presentation needs improvement, since the way of presentation is monotonous (e.g. Figure 3).  

Response2: We made some improvements to Figure 3 by adding standard deviations. The Figure 3 is mainly designed to show the differences between nutrients, and it contains the results of the one-way ANOVA (F and P vlues), as well as the multiple comparison test between different species. In addition, to show similar nutrient characteristics among some species, we visualized the results of the cluster analysis on the far left.

Point 3: In the whole paper, there is not any mention of the replicates used during sampling and analysis. These must be provided in materials and Methods section, with Figure and Tables where required.

Response3: We apologize for not expressing this clearly in the previous manuscript. We selected three adult plants in each species and then measured their soil, root, stem, and leaf indexes separately, and in fact they were treated as three replicates. So, we highlighted them as three replicates in the newly uploaded manuscript (line 111). In addition, the number of replicates used in the statistics is also indicated in Table 1.

Round 2

Reviewer 4 Report

The authors have sufficiently answered to my previous remarks; thus, I can recommend the ms for publication in its present form. Of course, the main problem remaining is the low originality, so the Editors are responsible to decide about this.